# An Empirical Mode Decomposition-Based Method to Identify Topologically Associated Domains from Chromatin Interactions

**Xuemin Zhao** [1] , **Ran Duan** [1] **and Shaowen Yao** [2,*]

1. School of Information Science and Engineering, Yunnan University, Kunming 650500, China; xmzhao@ynu.edu.cn (X.Z.); duanran@mail.ynu.edu.cn (R.D.)
2. Engineering Research Center of Cyberspace, Yunnan University, Kunming 650500, China
* Correspondence: yaosw@ynu.edu.cn; Tel.: +86-13629460312

**Abstract:** Topologically associated domains (TADs) represent essential units constituting chromatin's intricate three-dimensional spatial organization. TADs are stably present across cell types and species, and their influence on vital biological processes, such as gene expression, DNA replication, and chromosomal translocation, underscores their significance. Accordingly, the identification of TADs within the Hi-C interaction matrix is a key point in three-dimensional genomics. TADs manifest as contiguous blocks along the diagonal of the Hi-C interaction matrix, which are characterized by dense interactions within blocks and sparse interactions between blocks. An optimization method is proposed to enhance Hi-C interaction matrix data using the empirical mode decomposition method, which requires no prior knowledge and adaptively decomposes Hi-C data into a sum of multiple eigenmodal functions via exploiting the inherent characteristics of variations in the input Hi-C data. We identify TADs within the optimized data and compared the results with five commonly used TAD detection methods, namely the Directionality Index (DI), Interaction Isolation (IS), HiCKey, HiCDB, and TopDom. The results demonstrate the universality and efficiency of the proposed method, highlighting its potential as a valuable tool in TAD identification.

**Keywords:** Hi-C; topologically associated domains; empirical mode decomposition





## 1. Introduction

With progressive advancements in chromatin conformation capture technologies, researchers have successfully generated comprehensive three-dimensional interaction maps of all chromatins. These studies have gradually revealed the three-dimensional spatial organization of chromatin in the nucleus, covering different scales such as chromosome territories [1], A/B compartments [2] spanning approximately 5 Mb, topologically associated domains (TADs) [3] spanning several hundred Kbs, and smaller unified structural chromatin loops [4] at the genome-wide level. TADs serve as critical regulatory units for gene expression, facilitating chromatin interactions within specific regions while inhibiting interactions between different regions. Thus, TADs control the precise range of enhancer and promoter activities [5] at different scales of chromatin 3D architecture.

TADs have been observed to form within genomic regions spanning approximately 1 Mb in length. These TADs exhibit significantly stronger intra-domain chromatin spatial interactions than inter-domain chromatin spatial interactions. In particular, the spatial proximity between two points within TADs is closer than the distance between two points outside of TADs, even when considering the same linear distance. These structural domains have a significant impact on the regulation of biological functions. Dekker and Heard [6] extensively characterized TAD structures in different species, which can vary in size, formation mechanisms, and functional properties across organisms. TADs represent functionally distinct domains involved in gene regulation. Furthermore, the application of Hi-C technology in plants has revealed distinct TAD structures in genomic interaction

mapping studies conducted on rice, maize, tomato, sorghum, and cotton in recent years [7]. TADs serve as the primary functional units within three-dimensional chromatin architecture. The conservation of TAD structures has been observed across cell lines and even between species, suggesting that TADs maintain their structural integrity independent of cellular specificity. The boundaries of TADs show enrichment of chromatin-structuring proteins and correspond closely to the boundaries of replication units during biological processes. TADs are the fundamental units that constitute the three-dimensional spatial organization of chromatin, are consistently present in cells across species, and exert significant influence and regulation over essential biological processes, such as gene expression, DNA replication, and chromosomal translocation.

TADs manifest as regions characterized by increased internal interactions, which are visually represented as square patterns along the diagonal in Hi-C heat maps [8]. These square regions observed on thermograms vary in size and spacing, encompassing chromatin segments that have robust intra-regional interactions while exhibiting weaker interactions with neighboring chromatin segments. The boundaries of these square regions correspond to the boundaries of TADs. Effectively and accurately distinguishing the TAD structures from the vast amount of data present in Hi-C heat maps during three-dimensional chromatin structure analysis is a significant and complex challenge.

Based on Hi-C data, several methods have been developed to identify TADs within chromatin structures. Jesse et al. [3] examined TAD structures in the Hi-C interaction matrix. They proposed a method to characterize the deviation of chromatin regions from upstream and downstream interactions. This method uses the Directionality Index (DI) to characterize each chromatin region and allows for the inference of TAD positions using a hidden Markov model. Emily et al. [9] used the Interaction Isolation (IS) Index, which calculates the sum of interaction intensity values within local chromatin regions, to identify TADs. Through identifying boundaries according to interaction intensity, TAD boundaries can be determined via locating local extremes after performing a smoothing transformation. Both methods involve transforming the matrix information into a one-dimensional series and detecting TAD boundaries based on numerical changes.

The TopDom method [10] considers the relatively weak interactions between TAD boundaries and adjacent chromatin segments, and uses an adjustable window size to restrict the range of calculated interaction frequencies. Statistical features of the interaction frequencies within the window are extracted near the diagonal of the Hi-C interaction matrix, and the local minimum of the mean interaction strength is identified as a potential TAD boundary point. The Arrowhead method [4] applies a custom Arrowhead transformation to the interaction frequency matrix and calculates a custom score based on structural domain features. Custom scores above a certain threshold are considered potential structural domain vertices. The TADtree method [11] adopts a top-down strategy, which first identifies the outer structural domains using a method similar to the DI, and then the inner substructural domains. Conversely, the CaTCH method [12] adopts a bottom-up strategy through first using a small window to identify underlying structural domains, and gradually expanding the window to identify higher-level structural domains. The HiTAD method [13] further refines topologically related structural domains through globally optimizing chromatin partitioning and identifying hierarchically structured domains based on this optimization. The SuperTAD method [14] utilizes dynamic programming with polynomial time complexity to compute the coding tree of a Hi-C interaction map, with emphasis on minimal structural data. The TADBD method [15] employs a Haar diagonal template, a compact integrogram for acceleration, multi-scale aggregation for template size, statistical filtering, and alternate input/output options to detect TAD boundaries. Yan et al. [16] present MrTADFinder, an algorithm based on the network science notion of modularity, to detect TADs from intra-chromosomal contact maps. The CASPIAN method [17] utilizes a density-based hierarchical clustering algorithm, based on a distance metric, to cluster points within the Hi-C interaction matrix. Subsequently, the algorithm determines TAD boundaries according to the clustering results. Xing et al. [18] present a novel approach,

HiCKey, to discover hierarchical TAD structures in Hi-C data and conduct cross-sample comparisons. The HiCDB method [19] identifies TADs boundaries through constructing a local relative insulation (LRI) metric that converts a two-dimensional Hi-C map into a one-dimensional vector.

Chromatin interactions within the same TAD show more vigorous intensity than interactions between TADs. Distinct demarcation lines at TAD boundaries are often observed in the distribution of interactions within upstream or downstream chromatin regions. Researchers have compared different TAD identification methods at different resolutions using the same dataset and observed differences in metrics such as size, isolation, and sensitivity. This suggests the importance of selecting appropriate methods based on specific research objectives or using multiple methods to comprehensively understand Hi-C data [20–22]. More precise detection methods with improved accuracy and sensitivity are needed to detect TADs in high-dimensional, sparse, and noisy Hi-C data.

In the remaining sections of this paper, we describe our proposed fusion scheme in Section 2, including the motivation and discussion of parameter setting. We present the analysis and results in Section 3. Finally, the conclusions are drawn in Section 4.

## 2. Methods

### 2.1. Empirical Mode Decomposition Topologically Associated Domain

Huang et al. [23] introduced the empirical mode decomposition (EMD) method in their NASA research, which offers advantages of linearizing and smoothing nonlinear and non-smooth signal data without requiring any prior knowledge. This method adaptively decomposes complex data into a set of intrinsic mode functions (IMFs) based on the inherent characteristics of the input dataset itself. The Hilbert transform is then applied to each IMF to extract the instantaneous frequency and amplitude of each eigenmode function, thereby capturing their temporal variations. Through combining the information from all IMFs, a comprehensive time–frequency distribution of non-smooth signals can be obtained. EMD [23], originally developed for nonlinear and non-smooth signals, posits that each signal consists of distinct eigenmode functions, each of which exhibits potential linearity or nonlinearity. Subsequently, researchers have extended the application of EMD to various domains, including image processing [24–26].

TADs represent contiguous blocks along the diagonal of the Hi-C interaction matrix, which are characterized by dense intra-block interactions and sparse inter-block interactions. In this study, we proposed an innovative TAD identification method called the empirical mode decomposition topologically associated domain (EMTAD) method. The EMTAD method optimizes Hi-C interaction matrix data using EMD, adaptively decomposes Hi-C data into multiple eigenmode functions, and then reconstructs the Hi-C interaction matrix. Through incorporating the EMD technique, the EMTAD method effectively enhances the structural modes present in Hi-C data, reduces data noise, and provides more precise and more plausible TAD structures.

To identify TADs, we use Count Index (CI) [27,28], which is consistent with the original TAD definition. We compared the results of the EMTAD method with five conventional identification methods, namely the classical DI, IS, HiCKey, HiCDB, and TopDom. Our results show that the EMTAD method achieves the most favorable TAD identification results.

### 2.2. Data Preprocessing

The EMTAD method is a method designed to optimize Hi-C data and facilitate the identification of TADs. First, the Hi-C interaction matrix is normalized using the iterative correction and eigenvector decomposition (ICE) technique [29], as shown in Figure 1A. The original Hi-C interaction matrix exhibits significant noise, and regions away from the diagonal line contain significant interaction data similar to TAD structures. Such noisy data significantly affect the accuracy and reliability of TAD structure and downstream analyses.

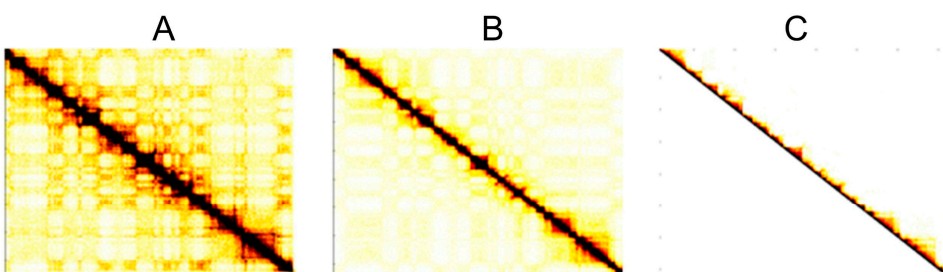

**Figure 1.** Heat map of the Hi-C interaction matrix at various treatment stages: (**A**) raw Hi-C interaction matrix; (**B**) ICE normalized interaction matrix; (**C**) fusion structure data interaction matrix. (Rao2014-GM12878-DpnII-allreps-filtered-50 kb).

Figure 1B illustrates the effectiveness of ICE normalization in removing noises from the original Hi-C interaction matrix. At the same time, it enhances the data surrounding TAD structures along the diagonal, thereby minimizing or correcting biases in the sequencing data and preventing error amplification. In addition, ICE normalization takes advantage of the symmetry of the Hi-C interaction matrix, allowing the method to operate only on the upper triangular portion of the matrix in subsequent steps. This reduction in computational overhead effectively reduces the memory requirements of the method.

Since TADs typically range in size from about 200 Kb to 2 Mb, interactions beyond the 2 Mb region can be ignored in the Hi-C interaction matrix. Furthermore, according to the definition of TADs, interaction values within TAD regions have higher intensity and lower variability. In contrast, interaction values in the surrounding regions between TADs have lower intensity and lower variability. Therefore, evaluating the proportion of change in interaction values and their absolute magnitudes, followed by applying logarithmic transformations to the matrix, makes it possible to effectively eliminate specific interaction values that persist within the inter-TAD regions. These interaction values correspond to biological interactions between two TADs and are not due to experimental errors. However, for the specific purpose of TAD identification using the EMTAD method, these inter-TAD interaction values provide unfavorable information and should be removed.

After applying ICE normalization, which includes removing interaction values beyond the 2 Mb region and excluding interaction values between TADs, the error inherent in the original Hi-C interaction matrix can be significantly reduced. At the same time, this process eliminates data that are not relevant for TAD detection. The result is the optimized Hi-C interaction matrix in Figure 1C that maximally preserves the TADs. This matrix serves as the basis for subsequent steps involving EMD of the Hi-C matrix and TAD identification.

### 2.3. Using Information Entropy to Measure IMF Matrix Values

The successful application of image decomposition and feature extraction of components using the EMD method in image processing provide valuable insights for identifying TADs within the Hi-C interaction matrix. In particular, the EMTAD method offers a convenient approach that minimizes the need for explicitly considering TAD structures, including TAD boundaries. Instead, the method adaptively decomposes complex, high-dimensional, sparse, and noisy Hi-C data into a set of eigenmode functions known as IMFs. This decomposition process merges the augmented Hi-C interaction matrix with the IMF set with the highest information entropy sum. Through iteratively evaluating the information entropy sum over all IMFs, the set of IMFs with the largest entropy sum is selected for fusion, resulting in an extended Hi-C interaction matrix.

In the context of TAD identification, the distinctive features of TADs manifest themselves as robust intra-domain interactions juxtaposed with weaker inter-domain interactions. Consequently, the boundary region between two TADs serves as a central reference for partitioning them into separate rectangular blocks. To effectively detect TAD boundaries, it is imperative to focus on information that deviates from the surrounding context within the contact matrix. To quantify the significance of the decomposed IMF matrix, the

concept of information entropy, as introduced in reference [30], is used. Higher information entropy values within the IMF matrix indicate greater variations in pixel values between neighboring regions, suggesting the presence of potential TAD boundaries.

The information entropy of a given *i*-th IMF ($imf_i(m)$) is formally defined as follows:

$$H_i = \sum_{m=1}^{M} p_{im} log p_{im} \tag{1}$$

where $p_{im} = \frac{imf_i(m)}{\sum_{m=1}^{M} imf_i(m)}$, $imf_i(m)(m = 1, 2, \ldots, M)$ is the *m*-th element of $imf_i$.

As can be seen from Equation (1), an increase in the uncertainty associated with the pixel values captured within an image corresponds to a proportional increase in the entropy value.

*2.4. Major Steps*

As shown in Figure 2, the EMTAD procedure includes the following key steps:

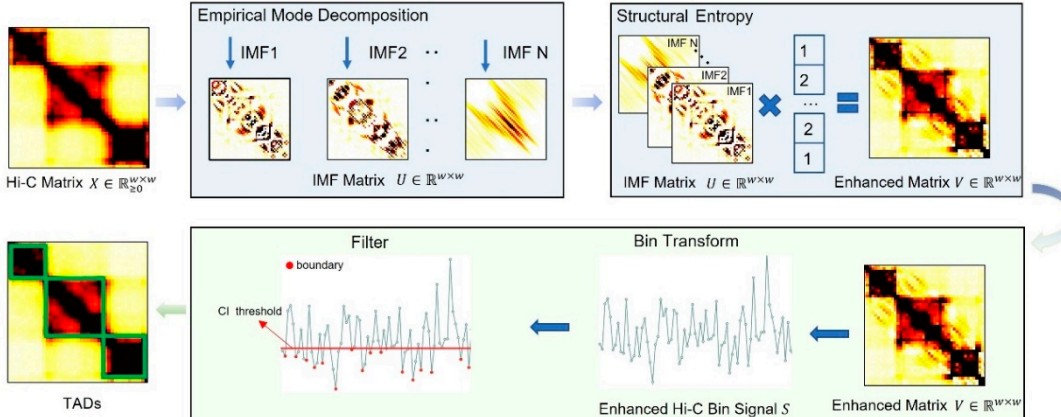

**Figure 2.** Framework diagram for the EMTAD methodology.

1. Normalize the input and remove irrelevant interaction values from the Hi-C contact matrix *U*.

2. Identify the local maximum points $u_{max}(t)$ and local minimum points $u_{min}(t)$ within the matrix *U* via selecting the maximum and minimum values within a $3 \times 3$ neighborhood.

3. Use the cubic spline functions to connect all extreme points $u_{max}(t)$ and form the upper envelope $e_{max}(t)$ of the matrix. Similarly, cubic spline functions are used to connect very small points $u_{min}(t)$ and form the lower envelope $e_{min}(t)$.

4. Calculate the mean of the upper and lower envelopes, denoted as $m(t) = \frac{[e_{max}(t) + e_{min}(t)]}{2}$.

5. Subtract the mean $m(t)$ from the matrix *U*.

6. Evaluate the iterative stopping condition for EMD, where the standard deviation SD $\leq \varepsilon$ (set $\varepsilon = 0.3$). If the condition is met, proceed to Step 7; otherwise, use the results from Step 5 and return to Step 3 for further calculation.

7. Obtain the L IMF components $mf_1 \sim imf_L$, followed by calculating their respective information entropy values using Equation (1). The information entropy values of the L IMF components are then summed.

8. Terminate the EMD process when steps 3 to 6 satisfy the iteration-stopping condition. The IMF components of all combinations and their corresponding sums of information entropy values are listed.

9. Determine the set of IMF components with the highest sum of information entropy values and generate the optimized Hi-C contact matrix *V* via superposition.

10. Identify the TADs within the Hi-C matrix *V* based on the CI values.

## 2.5. Algorithm

The following Algorithm 1 is the EMTAD algorithm:

---
**Algorithm 1:** EMTAD
---
Input: The initial Hi-C matrix is referred to as matrix $\boldsymbol{U}$.
Output: The enhanced and optimized Hi-C matrix $\boldsymbol{V}$.
Procedure:

1.  Import the Hi-C matrix $U$, initialize $i = 1$.
2.  Perform EMD on the matrix $U$, resulting in the derivation of $n$ IMFs ($imf_i(m)$).
3.  for $i = 1$ to $T$ do
4.  Calculate all local maxima points $u_{max}(t)$ and local minima points $u_{\min}(t)$ of the matrix $U(t)$.
5.  Use cubic spline functions to connect all identified local maxima points $u_{max}(t)$ to construct the upper envelope $e_{\max}(t)$ of the signal; similarly, use cubic spline functions to connect all identified local minima points $u_{\min}(t)$ to construct the lower envelope $e_{\min}(t)$ of the signal.
6.  Calculate the average of the upper and lower envelope curves $m(t) = \frac{[e_{\max}(t) + e_{\min}(t)]}{2}$.
7.  $h(t) = U(t) - m(t)$.
8.  Calculate the total information entropy $H_i$ for the IMFs by summing the probability distribution of each mode m within the set of IMFs using the formula $H_i = \sum_{m=1}^{M} p_{im} \log p_{im}$.
9.  end for
10. Select the combination of IMFs with the highest total information entropy, merge the n IMFs to obtain the enhanced and optimized Hi-C matrix $V$.
11. Evaluate the CI for each bin and record it in the $C_i List$.
12. for $C_i$ in $C_i List$ do
13. If the CI corresponds to a local minimum within the $C_i List$ and $C_i < C_t$ where $C_t$ denotes the threshold value of $C_i$
14. The current bin represents a boundary of TADs.
15. end for
---

## 2.6. Experimental Implementation Details

To validate the universality and stability of the EMTAD method, a series of experiments were performed using the EMTAD method to optimize and improve the Hi-C matrix dataset of five different cell lines, namely GM12878, HMEC, HUVEC, K562, and NHEK. The results of TAD identification were then compared with five alternative methods, namely the DI, IS, HiCKey, HiCDB and TopDom. The purpose of these comparative analyses was to evaluate the effectiveness and reliability of the EMTAD method.

The Hi-C data used in the experiments were obtained from the datasets published in 2014 by Rao et al. [4]. Specifically, the experiments predominantly used the Hi-C datasets (accession number GSE63525) available in the NCBI database, which included seven different resolutions (10 Kb, 25 Kb, 50 Kb, 100 Kb, 250 Kb, 500 Kb, and 1 Mb). These datasets were derived from five different cell lines, namely GM12878, HMEC, HUVEC, K562, and NHEK.

The ChIP-seq data used for comparative analyses of results included data on transcription factors and various histone modifications. These data were obtained from the database of the Encyclopedia of DNA Elements (ENCODE) project.

To assess the universality and stability of the EMTAD method, the obtained results were compared with the recognition results of two classical one-dimensional methods, namely the DI and IS. In addition, the widely used TopDom method was used for comparison due to its higher recognition accuracy. These comparative analyses were performed to validate the recognition efficiency of the EMTAD method.

The evaluation metrics were used to analyze the results, including the number of identified TADs, the boundaries delineating recognized TADs, the Silhouette Coefficient, the mutual information between the six methods (EMTAD, DI, IS, HiCKey, HiCDB, and TopDom), and the consistency of the recognition results. These metrics were used to evaluate the performance and reliability of different identification methods.

The code for the EMTAD method was developed in Python 3.7, while the EMD module was implemented using the PyEMD package. The experiments were performed on a computer system consisting of an Intel Xeon W-2123 3.6 GHz CPU, 64G DDR4 2666 RAM, and an NVIDIA GeForce RTX 2080Ti 11G graphics card. The operating system used for the experiments was Ubuntu 21.04.

## 3. Experimental Results

The EMTAD method was performed at seven resolutions (10 Kb, 25 Kb, 50 Kb, 100 Kb, 250 Kb, 500 Kb, and 1 Mb) in five different cell lines, namely GM12878, HMEC, HUVEC, K562, and NHEK. At the same time, the DI, IS, HiCKey, HiCDB, and TopDom methods were also applied. Since HiCKey and HiCDB are primarily used for identifying hierarchical TADs, we take out the outermost hierarchical TAD boundaries of HiCKey and HiCDB in this experiment for comparison with the recognition results of other methods. All different methods shared a common goal of detecting TAD boundaries and identifying TADs based on changes in interaction values within the interaction matrix.

The results showed that the EMTAD method performed better in identifying TADs compared to the other five methods across all seven resolutions in the five cell lines. However, minor variations were observed in some data sets, although the overall trend was consistent with this conclusion. In particular, the GM12878 dataset had higher data quality and a larger number of reads, making it more suitable for comprehensive data analysis.

### 3.1. Comprehensive Analysis and Comparisons of Similarity in TADs

In the experiments, six different methods for TAD identification were implemented in five different cell lines using Hi-C data with different resolutions. The results, shown in Figure 3A as an illustrative example, revealed that in the GM12878 cell line with a resolution of 50 Kb, the DI method showed the lowest TAD count, mainly at around 100, across chromosomes 1 to 22 and X. Conversely, the IS method showed a number of TAD counts, mainly in the range of 200 to 300, with a peak value reaching 700. The identification count number of the HiCDB method focuses on a range of 100 to 300, while the HiCKey method identifies approximately 300 to 400, with peak values exceeding 700. The TopDom and EMTAD methods yielded TAD counts primarily in the range between 100 and 400, with similar distributions. However, TopDom showed a higher peak value of 600. Notably, the EMTAD method showed higher accuracy in its identification results when comparing the TAD counts.

When comparing the existing methods for identifying TADs as documented in the literature [20–22], it became clear that the TADs identified using these methods exhibited inconsistencies in their number and length. Furthermore, the predicted TAD boundaries generated via these methods showed significant variability in their spatial localization. To address these issues, a nearest-neighbor voting approach was used to score the TAD boundaries identified through these different methods. Specifically, a TAD boundary was assigned a score of one if it was identified using a particular method within a given resolution interval. Through applying this scoring scheme, a cumulative score ranging from one to six was assigned to each TAD boundary, with a score of six indicating unanimous recognition via all six methods. This inter-comparison methodology was used to objectively evaluate the effectiveness of the different methods in identifying topologically related structural domains.

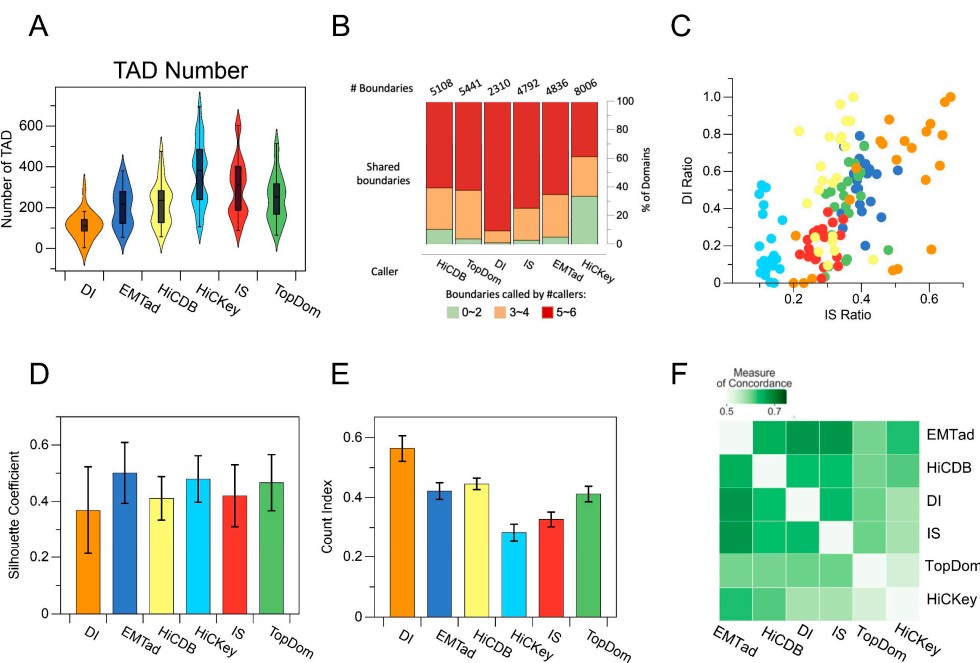

**Figure 3.** A comparative analysis of six different methods for the identification of TADs: (**A**) Evaluation of the total number of TADs and their distribution patterns. (**B**) Comparative evaluation of TAD boundary scores specifically within chromosome 2. (**C**) Investigation of the DI and IS coefficients associated with TAD boundaries. (**D**) Comparison of profile coefficients associated with TADs. (**E**) Evaluation of interaction coefficients associated with TADs. (**F**) Consistency comparison of identification results.

As shown in Figure 3B, when the results from Figures A and B are combined, the DI method detects fewer TAD boundaries. Although the DI method has higher boundary scores, implying improved accuracy, it may miss many authentic TAD boundaries. In contrast, the EMTAD method identifies more TAD boundaries while maintaining a high ratio of boundary scores (scores of four and three). When compared to the TopDom and IS methods, the EMTAD method effectively filters out low-scoring boundaries, which are predominantly indicative of potentially weak or false-positive boundaries. Based on these results, EMTAD demonstrates superior accuracy in identifying TAD boundaries while effectively mitigating false-positive boundary identifications.

The DI [3] method quantifies the deviation of a chromatin region's interactions from upstream or downstream regions. A DI value close to zero indicates a nearly equal frequency of interactions with upstream and downstream regions, suggesting its potential location at the boundary of a TAD. Similarly, the IS [9] method measures the cumulative strength of interactions within a given region. Analogous to the DI value, a higher IS value is generally observed within a TAD, while a region with a significantly low IS value may indicate a potential TAD boundary. Through normalizing the DI values to the IS values within the range (0, 1), a scatter plot was constructed to represent the average interaction values of these two metrics per chromosome. The proximity of the values in the scatter plot to the upper right corner indicates a closer fit of the average boundary metrics to the definition of the TAD boundaries. As shown in Figure 3C, specific scatter points of the DI, HiCDB, and HiCKey metric approach the upper left corner, indicating that the boundary identification results obtained using the DI, HiCDB, and HiCKey exhibit favorable behavior in specific chromosomal regions, but perform inadequately in the remaining chromatin segments. This observation highlights the partial instability of the TAD identification methods of the DI, HiCDB, and HiCKey. In contrast, the average index of boundaries identified using the EMTAD method shows a relatively higher concentration, reflecting the method's robustness across different chromosomes. In addition, the scatter values of

the boundaries identified using the EMTAD method are higher than the TopDom and IS methods, indicating a better agreement with the definition of TAD boundaries. Overall, the comparative analysis of the EMTAD method with the DI and IS methods highlights its effectiveness and robustness in accurately identifying TAD boundaries.

Silhouette coefficient [31] is a metric commonly used in traditional machine learning to evaluate the clustering results of clustering algorithms. In the context of TAD detection, each genomic bin is treated as a node and the interactions between bins are considered as edges, which are then used to construct a graph. Subsequently, identifying TADs can be viewed as a subgraph partitioning problem. This process can be likened to a clustering procedure, which allows the evaluation of the effectiveness of TAD detection using contour coefficients. Figure 3D shows that the DI method has a low mean and a large variance, indicating its inherent instability and suboptimal delineation performance. This result can be attributed to DI's limited ability to detect a sufficient number of TAD boundaries, resulting in the omission of certain true boundaries. From a clustering perspective, the DI method tends to merge clusters that should be separated, resulting in a lower score. In contrast, the EMTAD method has a higher contour coefficient score and lower variance than the other five methods, which is more consistent with the clustering perspective. Consequently, these results suggest that the EMTAD method achieves a superior quality regarding TAD recognition.

The CI [27,28] quantifies the amount of variability in the interactions between upstream and downstream regions of a chromatin region. It is determined by the ratio of the interactions within each upstream and downstream region to the interactions between these regions. In the case of a TAD boundary region, a higher CI value indicates greater variability. As shown in Figure 3E, the average effect of the Interaction Isolation Index (IS) is lower among the six methods examined. This suggests the presence of a potential margin of error between the results obtained from the IS method and the actual boundaries of the TADs. Conversely, the DI method shows the most favorable impact, albeit with a larger variance, highlighting the inherent instability of the DI method from the perspective of one-dimensional signals.

The methods of EMTAD and TopDom show superior performance in terms of impact, indicating the higher quality of TADs identified using these methods. However, the variance of the EMTAD method is smaller, indicating greater stability compared to the TopDom method.

To evaluate the consistency of TAD recognition results for a given pair of TADs, the Measure of Concordance (MoC) can be used. The MoC quantifies the degree of overlap between two TADs in terms of the number of base pairs, taking into account the overall size of each TAD [22]. The MoC is a normalized metric that ranges from zero (indicating complete lack of concordance) to one (indicating complete concordance), with higher values indicating greater concordance. In Figure 3F, a pairwise comparison was made between the recognition results obtained using the different methods. The EMTAD method showed the highest correlation with the results of the other methods. This observation highlights the significant overlap between the TADs identified using the EMTAD method and those identified using the other methods. Therefore, the TAD identification achieved using the EMTAD method can be considered reliable and comprehensive.

We also compared the accuracy of TAD boundaries identified via six different methods using simulated Hi-C data generated using machine learning methods. We first generate random TAD boundaries and then populate the matrix with identical random values within the defined boundaries. This process is iterated while introducing varying amounts of Gaussian noise into the matrix. As shown in Figure 4, we similarly used six methods to identify the tad boundaries of the simulated Hi-C data. The results show that the EMTAD method better detects TAD boundaries when comparing different scales of Gaussian noise.

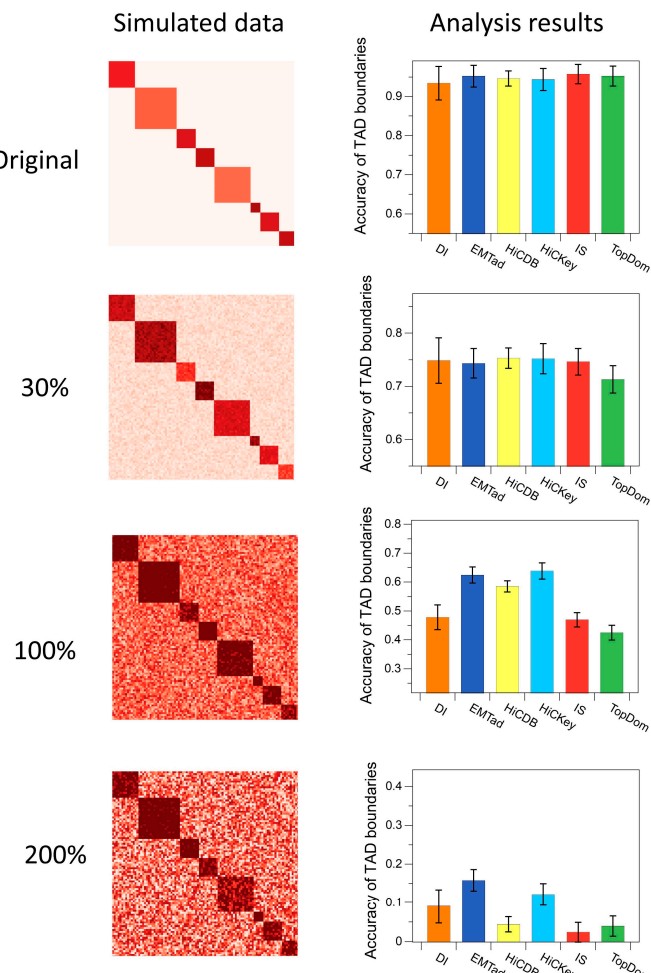

**Figure 4.** Results of simulated data analysis.

Several comparative analyses were performed, including comparisons of the total number of identified TADs, TAD boundaries, ratios of EMTAD-identified boundaries to DI and IS coefficients, means, and MI. These analyses collectively demonstrate that the EMTAD method performs better in accurately identifying TAD structures and boundaries than the other alternative methods. The results obtained using the EMTAD detection method consistently show higher stability and reliability, further validating its effectiveness.

### 3.2. Comparing Positions of TADs Identified Using Data at Different Resolutions

The information theory metric, mutual information (MI) [32], can be used to quantify the similarity between two clustering results. A higher MI value indicates a higher similarity between two clustering results. The MI value can exhibit the consistency of results through measuring the similarity between identification results of two TADs. Multiple experiments were performed using six different methods on Hi-C data, which were obtained from five cell lines, namely GM12878, HMEC, HUVEC, K562, and NHEK. The experimental setup is shown in Figure 5A. Comparative analyses were performed on the GM12878 dataset across 22 cell lines as well as on the X chromosome. Notably, when considering the results obtained from data at resolutions of 25 Kb and 50 Kb, at resolutions of 25 Kb and 100 Kb, and at resolutions of 50 Kb and 100 Kb, the pairwise comparisons consistently showed the highest degree of similarity in the TADs identified using the EMTAD method.

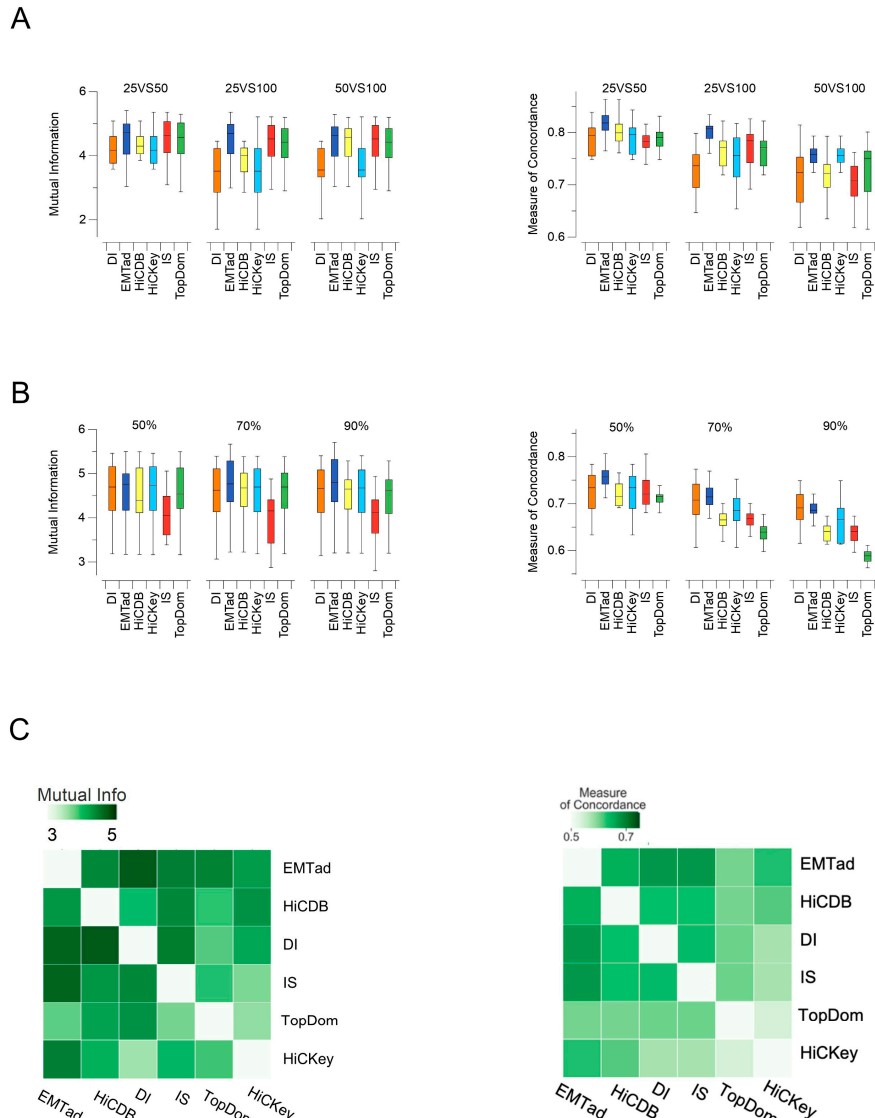

**Figure 5.** Comparison the positions of TADs identified using data at different resolutions: (**A**) consistency comparison of the six methods at 25 Kb vs. 50 Kb resolution, 25 Kb vs. 100 Kb resolution, and 50 Kb vs. 100 Kb resolution; (**B**) consistency comparison at different sampling depths; (**C**) consistency comparison at different resolutions and different sampling depths.

In this experiment, the six methods were compared individually using different sampling depths of 90%, 70%, and 50%. As shown in Figure 5B, the EMTAD method shows superior validity and consistency in TAD detection using the same comparative indicators.

In conclusion, the metrics mentioned above collectively demonstrate the consistent and superior boundary detection capabilities of the EMTAD method across different data resolutions. Furthermore, the EMTAD method shows remarkable robustness when confronted with different data resolutions.

Pairwise comparisons were performed at different resolutions, including 25 Kb vs. 50 Kb, 25 Kb vs. 100 Kb, and 50 Kb vs. 100 Kb. Pairwise comparisons were also performed using different sampling depths of 90%, 70%, and 50%. As shown in Figure 5C, the results of the EMTAD method show a significant correlation with the results of the other five methods. This correlation suggests that the EMTAD method possesses robust generalization and stability, and thus demonstrates superior robustness.

The comparative analyses of the recognition results of Hi-C data at different resolutions shows that the EMTAD method consistently outperforms the other five methods, yielding

slightly higher recognition accuracy. This finding underscores the remarkable adaptability of the EMTAD method to different resolution settings, as well as its superior generalization and robustness. In addition, the EMTAD method shows improved accuracy in identifying TADs, further proving its effectiveness.

### 3.3. Enrichment Analysis of Transcription Factors and Histone Modifications

The boundaries of TADs show a remarkable enrichment of transcription factors, including CTCF, RAD21, and SMC3, as well as histone modifications, such as H3K27ac, H3K4me3, and H3K36me3. These molecular components play critical roles in various biological processes and exert regulatory effects on DNA expression [3]. To evaluate the efficacy of the EMTAD method in accurately identifying TAD boundaries, we used the TAD boundaries and TADs detected at the resolution of 50 Kb in the GM12878 cell line as an illustrative example. Specifically, we examined the presence of three transcription factors (CTCF, RAD21, and SMC3) at TAD boundaries while also assessing the enrichment of histone modifications (H3K27ac, H3K4me2, H3K4me1, H3K4me3, H3K36me3, H3K79me2, H3K9ac, H3K9me3, H3K27me3, and H4K20me1) within TADs. Comparative analyses using different methods were performed to evaluate the enrichment patterns of transcription factors and histone modifications at TAD boundaries and within TADs.

Figure 6A–C show the enriched presence of three transcription factors, namely CTCF, RAD21, and SMC3, at the boundaries of TADs identified using different methods. Notably, the average enrichment of the transcription factors at the boundaries of TADs recognized using the EMTAD method exceeded that of the other five methods. In addition, the boundaries identified using the EMTAD and DI methods showed more concentrated average enrichment curves within the central regions of the boundaries, indicating their improved accuracy. In contrast, the boundaries identified using the TopDom and IS methods showed potential dynamics, resulting in lower peak ratios in their interaction curves.

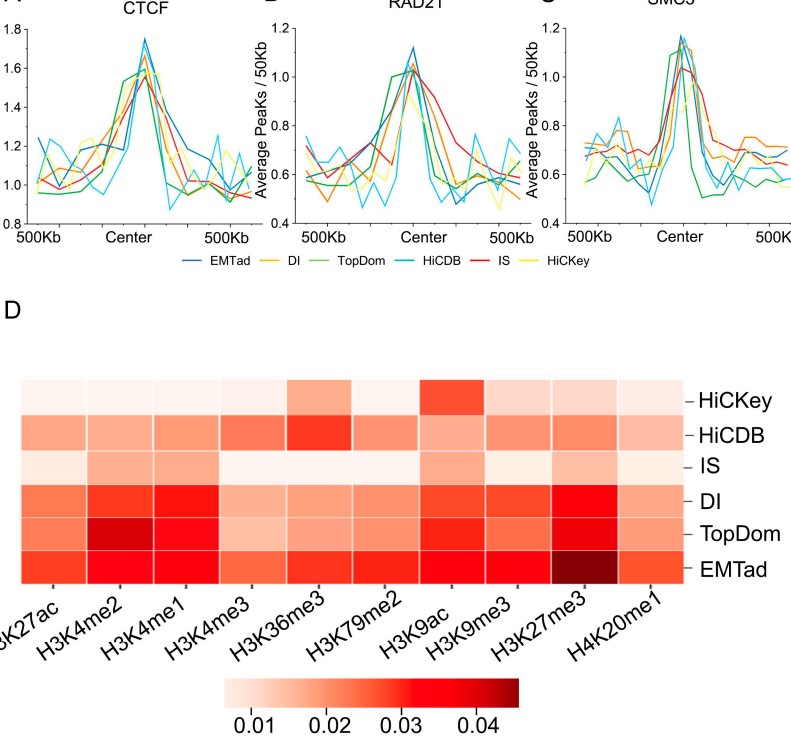

**Figure 6.** Comparison of results obtained using different methods for evaluating enrichment of TAD boundary transcription factors and TAD regional histone modifications: (**A**) comparison of CTCF enrichment analysis; (**B**) comparison of RAD21 enrichment analysis; (**C**) comparison of SMC3 enrichment analysis; (**D**) comparison of histone enrichment analysis of TAD regions.

A comparative analysis of histone modification enrichment within TADs identified via the six methods was performed. As shown in Figure 6D, the IS and HiCKey methods showed the lower enrichment of histone modifications within TADs. This observation can be attributed to their identification of the highest number of TADs, as shown in Figure 3A. Conversely, the remaining four methods showed more consistent enrichment patterns of histone modifications within TADs. Among the ten histone modifications examined, the EMTAD method showed higher enrichment than the other five methods for the majority of histone modifications, except for the TopDom method, which showed the highest enrichment for H3K4me2.

An analysis of enrichment for transcription factors and histone modifications within the identified boundaries and regions of TADs revealed that TADs and TAD boundaries predicted using the EMTAD method have specific biological functions. Thus, the EMTAD method demonstrates superior accuracy in detecting TAD structures and boundaries compared to the other methods, with slightly higher enrichment levels observed in the identified TAD boundaries and regions. As a result, the TAD detection results obtained using the EMTAD method are considered more accurate and reliable.

## 4. Conclusions

We propose a TAD identification method EMTAD, based on EMD which includes several steps: (1) normalizing the original Hi-C interaction matrix with ICE to account for biases; (2) removing interaction values larger than 2 Mb and interactions between TADs; (3) applying EMD to the Hi-C interaction matrix, resulting in N IMFs; (4) calculating the information entropy for each IMF and identifying the IMF component with the highest sum of information entropy; (5) reconstructing the matrix using the selected IMF component to obtain an optimized and improved Hi-C interaction matrix; and (6) performing TAD identification based on the reconstructed matrix. The source code is available online through GitHub (https://github.com/ZhaoXuemin/EMDTAD accessed on 10 August 2023).

The results of our EMTAD method were rigorously compared with five established TAD identification methods, namely DI, IS, HiCKey, HiCDB, and TopDom. The comparison highlights the superior performance of the EMTAD method in terms of enhanced generalization, stability, and robustness in identifying TAD structures. Considering that TADs are recognized as crucial regulatory units within the genome, comprising finer sub-TADs such as chromatin loops and remote interactions, the accurate identification of TADs poses paramount importance in studying three-dimensional chromatin structures and gene transcriptional regulation.

**Author Contributions:** Conceptualization, X.Z. and R.D.; methodology, X.Z. and R.D.; software, R.D.; validation, X.Z. and R.D.; formal analysis, X.Z. and R.D.; investigation, R.D.; resources, R.D.; data curation, X.Z. and R.D.; writing—original draft preparation, X.Z. and R.D.; writing—review and editing, X.Z. and R.D; visualization, X.Z. and R.D.; supervision, S.Y.; project administration, S.Y.; funding acquisition, S.Y. All authors have read and agreed to the published version of the manuscript.

**Funding:** This research was funded by the National Natural Science Foundation of China (Grant No. 61863036 and 72164037).

**Data Availability Statement:** The datasets used in this paper are publicly available and their links are provided in the reference section.

**Conflicts of Interest:** The authors declare no conflict of interest.

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
