# Peer review of "An Empirical Mode Decomposition-Based Method to Identify Topologically Associated Domains from Chromatin Interactions"

_electronics, doi:10.3390/electronics12194154_

Round 1

Reviewer 1 Report

The authors introduced a new method to detect linear TADs structures from Hi-C data matrices by using matrix decomposition techniques. Detecting TADs, especially hierarchical TADs is challenging in bioinformatics analysis of 3D chromatin interactions. The results seem to be promising, however, many concerns/questions are rising.

Major comments:

1. As a methodology paper, the code/pipeline should be released. I can't check the code and data to evaluate the results. Please release it publicly thorough GitHub or similar repositories. A archived version of the code and preprocessed data should be uploaded to DOMO website for reproducibility.

2. The description of method EMTAD should be improved. Although authors briefly mentioned the major steps of EMTAD, more technical details of each step are expected. For example, EMD method/package.

3. Many new methods are available for TAD detection, even for hierarchical TAD organizations. Can EMTAD detect hierarchical TADs?

4. Authors compared only with for three old methods (even TopDom in 2016). However, there are many new methods available. For example, SuperTAD, CASPIAN, TADBD, HicKey, MrTADFinder. Authors can find methods in review papers (PMID: 35104223, 35413815, 34415530). Authors should compare EMTAD with these new methods.

5. EMTAD is compared with other methods on only real Hi-C datasets by using predicted TADs from other methods (published results). It is not a good comparison as those predictions are not true positives. There are artificial datasets available for such comparison in above review papers or method papers, such as HiCKey. Authors should compare EMTAD with those methods by using artificial datasets and same criteria for performance evaluation.

6. Authors should introduce those new methods in the Introduction section.  

N/A

Author Response

On behalf of all the co-authors, I would like to thank you for your time spend making your constructive remarks and useful suggestions, which has significantly raised the quality of our manuscript and has enable us to improve the manuscript. each of your comment has been accurately revised. We gratefully appreciate for your valuable and detail comments. All the changes are marked in red color in this version. Detailed responses to your comments and suggestions are given below.

Reviewer 2 Report

Inspired by previous EMD method, the authors developed a new method named EMTAD to analyze TADs of Hi-C data. The approach is novel and may become a useful tool for 3-D genomics analysis. However, due to poor writing, the manuscript needs significant improvement.

Major changes:

1)      Organization: a) Section 2. Related work could be merged with Section 1. Introduction; b) Section 4.1. Experiments should be moved to Section3. Methods; c) A section of Discussion will be beneficial for advantages and disadvantages of EMTAD method.

2)      Strictly, Introduction is for instruction, not for discussion of results (e.g., Lines 70-75); Methods is for the detail of methods, with development strategy, not for discussion of results (e.g., Line 144); and Results is for the results; all the discussions could be moved to a new section, so as to clarify the manuscript structure.

3)      Please define the terms, such as accuracy, reliability, efficiency, and give the evidence. To me, Figure 5 is more about sensitivity improvement rather than accuracy improvement. As time is an important factor for HiC big data, the authors may compare the data processing time as well.

4)      Availability of algorithm.

Minor changes:

1)      Abbreviations: 1st time appearance with full name (e.g., TADs Line 33 requires full name), afterwards no full name any more (e.g., Lines 81-82, Line 132, etc.). This applies to all the others, such as DI, IS, CI, ICE, etc.

2)      Consistency: Lines 212-235. Step 1. Normalize…; Step 2. Identify …; Step 3. Use …; Step 4. Calculate …; …; Step 10. Identify …; last names with initial of first names in References; etc.

3)      Grammar throughout the manuscript, including Abstract, e.g., “the results are compared with those from three commonly used TAD detection methods,”

4)      Conciseness: e.g. “results” instead of “experimental results”; and abundant contents throughout the manuscript.

Please refer to the comments above.

Author Response

On behalf of all the co-authors, I would like to thank you for your time spend making your constructive remarks and useful suggestions, which has significantly raised the quality of our manuscript and has enable us to improve the manuscript. each of your comment has been accurately revised. We gratefully appreciate for your valuable and detail comments. We also uploaded our point-by-point response to the comments, and a detailed account on the changes that we have made in improving the quality of our manuscript. All the changes are marked in red color in this version. Detailed responses to your comments and suggestions are given below.

Round 2

Reviewer 1 Report

Thank the authors for addressing my concerns and questions. In the revision, the authors uploaded their code to GitHub and reviewed some newly published methods. However, there are still several questions that haven't been fully addressed. It is odd that the new results of comparisons among methods were not updated in the revision. It is unacceptable for there to be no further updating, comparison, and new experimental analysis due to time constraints.

Major comments:

(1) The authors partially responded to my question 1. The authors uploaded their code to GitHub, but there was no data and code backup to DOMO. On the method homepage on GitHub, the code and examples contain minimal information. More details are strongly needed to help readers/users understand their data and code. For example, their outputted file is just a txt file with columns presenting coordinates. But what are the biological details? Which chromosome? Which data example? Authors should include and update this information to make the website more informative.

Meanwhile, it is suggested that all the datasets used/analyzed and practical code for comparative analysis with other methods be archived in DOMO for further reproduction.

(2) Two methods, HICDB and HiCKey, were compared, but the results were not updated in the revision. Authors should update their figures and descriptions to provide clear results about the comparison of these methods.

(3) The Methods section needs further updating. Although minimal information has been added for these methodological steps, more experimental details are required. For example, what criteria were used in comparing different methods? It is not trivial to compare precision among different methods. The authors should use commonly accepted criteria to compare the performance among different methods.

(4) Authors mentioned that they compared different methods on artificial datasets, but there is no description of those datasets and results. This is unacceptable and must be improved.

Many grammar errors and overall presentation to be improved. A comprehensive editing from a native speaker is strongly suggested.

Author Response

Response to Editor and Reviewers

On behalf of all the co-authors, I would like to thank the Editor and all anonymous reviewers for their constructive comments and suggestions on this submission. These comments are very valuable for revising and improving our manuscript. We have addressed the reviewers’ comments and revised our manuscript carefully. We have now worked on both language and readability and have also involved native English speakers for language corrections. All the changes are marked in red in this version. Some compared results are included in the Supplementary Materials after the 4. Conclusions (Line480). We also uploaded our point-by-point response to the comments, and a detailed account on the changes that we have made in improving the quality of our manuscript. Detailed responses to these comments are given below.

Round 3

Reviewer 1 Report

Thank the authors for further improving their manuscript. In the current revision, they (1) added more code and results to GitHub, (2) added the comparison results as a supplementary file. However, there are no response/improvements for other concerns.

Major comments:

(1) The update to GitHub is minimal. I suggested to include more descriptions about the data input and output to help users to understand and use, but no updates. Although they provided Jupyter Notebooks for their anlysis, which is nice, but insufficient.

(2) Although the authors included the comparison with other two methods as supplementary file, it is suggested to be included in main text to demonstrate a whole picture of their performance to users/readers.

(3) Again, the authors should use simulation data to precisely benchmark their methods as well as other methods. If those datasets in HiCKey are not suitable, authors should find other simulation datasets used in other papers for the benchmarking. I would suggest to extend the due day if more time needed. 

(4) The authors claimed have improved their English editing, however, it seems there are no difference between the revision version 1 and 2. It seems only adding supplementary information in the current version. The certificate provided is marked for August-2023, could be an early version before authors further modifications. 

English editing to be improved.

Author Response

Detailed Response to Reviewers

On behalf of all the co-authors, I would like to thank the Editor and all anonymous reviewers for their constructive comments and suggestions on this submission. These comments are very valuable for revising and improving our manuscript. We have addressed the reviewers’ comments and revised our manuscript carefully. We have now worked on both language and readability and have also involved native English speakers for language corrections. All changes are marked in red in this version with respect to the first version. Some compared results are included in the main text (such as Figure 3). We also uploaded our point-by-point response to the comments, and a detailed account on the changes that we have made in improving the quality of our manuscript. Detailed responses to these comments are given below.

Round 4

Reviewer 1 Report

Thanks to the authors for their further updates to the manuscript, addressing my concerns, which mainly involved the comparison analysis of six methods in Section 3.1. Despite their efforts and updates, several questions still remain.

(1) While the comparison of six methods was included in Section 3.1, there is no comparison and updated results provided in Section 3.2. This should be performed and included.

(2) There has been no comparison performed using simulation data. The reason given, 'We are not familiar with the R language,' is deemed unacceptable for scientific research.

(3) GitHub updates have been minimal, with only 6 additions and 6 deletions recorded in README.md. I had previously suggested uploading their data and EMDTAD code, as well as code for comparing with other methods, to Zenodo https://zenodo.org/. The authors have ignored this suggestion but should upload to enhance the reproducibility.

Several minor grammar questions to be improved.

Author Response

Detailed Response to Reviewers

On behalf of all the co-authors, I would like to thank the anonymous reviewers for their constructive comments and suggestions on this submission. These comments are very valuable for revising and improving our manuscript. We have addressed the reviewers’ comments and revised our manuscript carefully. All changes are marked in red in this version with respect to the first version. Some compared results are included in the main text (such as Figure 4 , Figure 5 and Figure 6). We also uploaded our point-by-point response to the comments, and a detailed account on the changes that we have made in improving the quality of our manuscript. Detailed responses to these comments are given below.
